# Pro-Apoptotic Potential of *Pseudevernia furfuracea* (L.) *Zopf* Extract and Isolated Physodic Acid in Acute Lymphoblastic Leukemia Model In Vitro

**DOI:** 10.3390/pharmaceutics13122173

**Published:** 2021-12-16

**Authors:** Martin Kello, Tomas Kuruc, Klaudia Petrova, Michal Goga, Zuzana Michalova, Matus Coma, Dajana Rucova, Jan Mojzis

**Affiliations:** 1Department of Pharmacology, Faculty of Medicine, Pavol Jozef Šafárik University, 040 01 Košice, Slovakia; tomaskuruc@centrum.sk (T.K.); claudy.petrova@gmail.com (K.P.); zuzana.michalova@upjs.sk (Z.M.); matus.coma@upjs.sk (M.C.); 2Department of Botany, Institute of Biology and Ecology, Faculty of Science, Pavol Jozef Šafárik University, 041 67 Košice, Slovakia; michal.goga@univie.ac.at (M.G.); dajana.rucova@upjs.sk (D.R.)

**Keywords:** acute lymphoblastic leukemia, *Pseudevernia furfuracea*, physodic acid, apoptosis, oxidative stress, DNA damage, cell cycle checkpoints, MAPK

## Abstract

Acute lymphoblastic leukemia (ALL) is the most frequently diagnosed type of leukemia among children. Although chemotherapy is a common treatment for cancer, it has a wide range of serious side effects, including myelo- and immunosuppression, hepatotoxicity and neurotoxicity. Combination therapies using natural substances are widely recommended to attenuate the adverse effects of chemotherapy. The aim of the present study was to investigate the anti-leukemic potential of extract from the lichen *Pseudevernia furfuracea* (L.) *Zopf* (PSE) and isolated physodic acid (Phy) in an in vitro ALL model. A screening assay, flow cytometry and Western blotting were used to analyze apoptosis occurrence, oxidative stress, DNA damage and stress/survival/apoptotic pathway modulation induced by the tested substances in Jurkat cells. We demonstrate for the first time that PSE and Phy treatment-induced intrinsic caspase-dependent cell death was associated with increased oxidative stress, DNA damage and cell cycle arrest with the activation of cell cycle checkpoint proteins p53, p21 and p27 and stress/survival kinases p38 MAPK, JNK and PI3K/Akt. Moreover, using peripheral T lymphocytes, we confirmed that PSE and Phy treatment caused minimal cytotoxicity in normal cells, and therefore, these naturally occurring lichen secondary metabolites could be promising substances for ALL therapy.

## 1. Introduction

Acute lymphoblastic leukemia (ALL) is an aggressive disease and the most common type of childhood cancer. According to the National Cancer Institute [1], ALL represents 25% of cancer diagnoses among children under the age of 15. In the past decade, survival rates have approached 85%, mainly due to modern risk-based scoring and subsequent therapy. Standard treatment options for newly diagnosed ALL include chemotherapy (vincristine, doxorubicin, corticosteroids, L-asparaginase), radiotherapy and stem cell transplantation. Despite advances in B-cell therapy, which includes the bispecific antibody blinatumomab, antibody–drug conjugate inotuzumab ozogamicin and chimeric antigen receptor (CAR) T-cell therapy directed against the B-cell antigen CD19, only limited advances have been made in the treatment of T-cell ALL (T-ALL) [2,3,4]. Moreover, especially during the induction phase of T-ALL therapy, serious toxicities have been observed, leading to neutropenia and infection [5,6]. Vincristine, as the key component of chemotherapy, contributes to higher survival rates of patients with ALL. However, the use of vincristine is associated with side effects, including neurotoxicity [7]. Therefore, the identification of novel natural or synthetic substances with reduced toxicity and anti-cancer activity for ALL treatment is a major challenge in this research area [8,9,10,11,12,13].

The pharmaceutical potential of lichens and their secondary metabolites has recently attracted increasing attention worldwide, as reviewed in [14,15]. Many in vitro and in vivo studies have described the cytotoxic, anti-proliferative and anti-cancer effects of lichen-derived compounds, such as apoptosis induction, autophagy, cell survival and proliferation signaling pathway alteration, invasion, migration, angiogenesis and cellular senescence [16,17]. These published data indicate that different lichen metabolites or extracts act through different signaling pathways in various cell types. It is known that several extracts and active compounds (lobaric acid, protolichesterinic acid, usnic acid, leucotylic acid, cyaneodimycin) from the lichen genera *Cladonia*, *Stereocaulon*, *Lobariella*, *Myelochroa*, *Cornicularia* and *Lichina* possess anti-leukemic potential, as reviewed in Mohamadi et al. [18]. In this review, in addition to Jurkat cells, the potential anti-proliferative activity (MTT, BrdU assays) was investigated and summarized for some other leukemia cell lines, such as promyelocytic leukemia (HL-60), chronic myelogenous leukemia (K-562) and murine leukemia (L1210). Moreover, the effects of atranorin, the most studied metabolite among all secondary metabolites, were investigated in leukemia cells (Jurkat, HL-60). Backorova et al. [19] found that atranorin inhibited cell proliferation, induced apoptosis and initiated the accumulation of leukemia cells in the S phase of the cell cycle. Long ago, in 1959, strong anti-tumor activities of polyporic acid in mice inoculated with ALL were described by Burton and Cain [20]. Physodic acid was tested by a screening method on a chronic myelogenous leukemia (K-562) cell line, showing the cytotoxic potential and inhibition properties against M-phase phosphoprotein 1 [21]. However, despite the published data, very little is known about the molecular mechanisms and signaling pathways that are activated or modulated by the above-mentioned lichen extracts or secondary metabolites in ALL treatments.

The aim of the present study was to assess the anti-cancer potential of extract from the lichen *Pseudevernia furfuracea* (L.) *Zopf* and the metabolite physodic acid using the Jurkat cell line as an in vitro ALL model. The composition of the most abundant metabolites in *P. furfuracea* extract (PSE) is atranorin and chloroatranorine (depside group) and physodic acid, oxyphysodic acid and O-methylphysodic acid (depsidone group) [22], which together account for their overall anti-tumor effects through their interactions (synergy, additivity, antagonism, inhibition). To the best of our knowledge, published data on the anti-leukemic potential and cell death mechanisms mediated by PSE and physodic acid (Phy) do not exist. On the other hand, Phy or PSE possesses cytotoxic, pro-apoptotic, genotoxic and anti-migratory properties in a broad spectrum of cancer cell lines from different tissue origins, including colon, liver, breast, lung and skin [23,24,25,26,27]. Moreover, our latest paper described the effect of PSE and Phy on tumor microenvironment modulation, focusing on epithelial–mesenchymal transition (EMT), cancer-associated fibroblast (CAF) transformation and angiogenesis. We demonstrated that both PSE and Phy were able to modulate N-cadherin, fibronectin, α-SMA expression, or Slug and Smad2/3 signaling pathways in a concentration- and time-dependent manner [28]. On the basis of published data, we focused on the anti-proliferative effect, induction of apoptosis, and modulation of the signaling pathways mediated by extract from the lichen *P. furfuracea* and physodic acid in an in vitro acute lymphoblastic leukemia model.

## 2. Materials and Methods

### 2.1. Collection, Identification, Isolation and Characterization

Dr. Goga (September 2019) collected lichen *Pseudevernia furfuracea* (L.) Zopf from barks of *Picea abies* at Kojšovská hol’a (48.781049, 20.978567) in Volovské vrchy (Košice, Slovakia). The lichen specimen was identified and deposited in the herbarium of P.J. Šafárik in Košice (KO35800) by botanists and lichenologists Prof. Backor and Dr. Goga. PSE was prepared by acetone extraction and characterized by HPLC and NMR spectroscopy as described in Petrova et al. [28]. Lichen extract composition analyses showed that *P. furfuracea* (L.) *Zopf* primarily contains atranorin, chloratranorin (cortex) and physodic acid (medulla). After extraction and isolation, both PSE and Phy were dissolved in dimethyl sulfoxide (DMSO, final c ≤ 0.02%) and diluted directly in cultured medium. DMSO itself exhibited no cytotoxicity in cultured cells.

### 2.2. Cell Culture

The human T-lymphoblastic leukemia cell line Jurkat (ATCC, Manassas, VA, USA) and human T lymphocytes (HTLs, from blood isolation) were maintained in RPMI 1640 medium (Biosera, Kansas City, MO, USA). The growth medium was supplemented with 10% FBS (Thermo Scientific, Rockford, IL, USA) and antibiotic/antimycotic solution (Merck, Darmstadt, Germany), and cells were cultivated in an atmosphere containing 5% CO_2_ in humidified air at 37 °C. Before each experiment, the cell viability was estimated by trypan blue exclusion (≥95%).

### 2.3. Human Peripheral Blood Samples and T-Lymphocyte Isolation

Human peripheral blood samples were obtained from healthy volunteer (non-smoker, non-alcoholic, not under drug therapy, and with no recent history of exposure to mutagens) in the hematology department of L. Pasteur University Hospital in Košice. Blood samples were drawn with EDTA Vacutainer tubes (BD Biosciences, Franklin Lakes, CA, USA) and processed within 1 hour of collection. Peripheral blood mononuclear cells (PBMCs) were isolated by density gradient centrifugation using Histopaque-1077 (Merck). T cells were isolated from human PBMCs using Human T-cell Isolation Kit (StemCell Technologies, Vancouver, BC, Canada) according to the datasheet of the isolation kit. The purity of T cells was greater than 90%, as confirmed by flow cytometry.

### 2.4. Cell Viability Assay

The resazurin assay was used to determine the anti-proliferative/cytotoxic effect of PSE (concentrations of 10, 50, 100 µg/mL) and Phy (concentrations of 10, 50, 100 µM) on Jurkat and HTLs cells. Tested cells (1 × 10^4^/well) were seeded in 96-well plates. After 24 h, final PSE and Phy concentrations prepared from DMSO stock solution were added, and incubation proceeded for the next 72 h. Ten microliters of resazurin solution (Merck) at a final concentration of 40 µM was added to each well at the end-point (72 h). After a minimum of 1 h incubation, the fluorescence of the metabolic product resorufin was measured by the automated Cytation^TM^ 3 cell imaging multi-mode reader (Biotek, Winooski, VT, USA) at 560 nm excitation/590 nm emission filter. The results were expressed as a fold of the control, where control fluorescence was taken as 100%. All experiments were performed in triplicate. The IC_50_ values were calculated from these data.

### 2.5. Flow Cytometric Analyses

According to the experimental scheme, Jurkat or HTL cells were seeded in Petri dishes with complete growth medium and cultivated for 24 h. After 24 h, cells were treated with PSE (45 µg/mL) and Phy (60 µM) at IC_50_ concentrations. At the experimental end-point (3, 24, 48 or 72 h), adherent and floating cells were harvested together and washed in PBS. The cell suspension was divided for subsequent analysis, stained with un-/conjugated antibody or dye (Table 1) and incubated for 20 min (room temperature, in the dark). Then, if necessary, conjugated secondary antibody staining was carried out for 20 min. After incubation, the BD FACSCalibur flow cytometer (BD Biosciences, San Jose, CA, USA) was used for fluorescence detection. A minimum of 1 × 10^4^ events were analyzed per sample, and all experiments were performed in triplicate.

### 2.6. Cell Proliferation Analysis

To analyze the proliferation activity of Jurkat and HTLs cells, the CellTrace^TM^ Yellow Cell Proliferation Kit for flow cytometry was used (Thermo Scientific). The cells (1 × 10^6^) were resuspended in 1 mL of staining solution with a final concentration of 10 µM and incubated for 20 min at 37 °C in the dark. After incubation, cells were washed in a complete culture medium and incubated again for 5 min at 37 °C. After the second incubation, supernatant was removed by centrifugation, and the pellet was resuspended in fresh, pre-warmed complete culture medium and seeded in Petri dishes for 24 h. Afterwards, seeded cells were treated with PSE (45 µg/mL) and Phy (60 µM) and analyzed at three different time points (24, 48 and 72 h). For analysis of stained cells, a BD FACSCalibur flow cytometer (BD Biosciences) was used. A minimum of 1 × 10^4^ events were analyzed per analysis, and all experiments were performed in triplicate.

### 2.7. Cell Cycle Analysis

For the cell cycle analyses, Jurkat and HTL cells were harvested at three different time points (24, 48 or 72 h) after treatment (PSE 45 µg/mL and Phy 60 µM), washed in cold PBS and fixed in cold ethanol (70%). Samples were then stored at −20 °C until analysis (minimum of 24 h). Prior to each analysis, cells were stained with solution containing 0.5 mg/mL ribonuclease A, 0.2% final concentration of Triton X-100 and 0.025 mg/mL propidium iodide in 500 µL of PBS (all Merck) and incubated for 30 min at room temperature in the dark. The BD FACSCalibur flow cytometer (BD Biosciences) was used to analyze stained cells. A minimum of 1 × 10^4^ events were analyzed, and all experiments were performed in triplicate.

### 2.8. Annexin V/PI Staining for Phosphatidyl Serine Externalization

The phospholipid phosphatidyl serine (PS) was detected by the Annexin V–Alexa 633 conjugate after its externalization on the other side of the plasmatic membrane, which acts as a marker of programmed cell death. For apoptosis detection, Jurkat and HTL cells (1 × 10^6^) were harvested 3, 24, 48 or 72 h after PSE 45 µg/mL and Phy 60 µM treatment. NAC/PSE or NAC/Phy experimental groups were pre-treated with N-acetyl-L-cysteine (NAC c = 2 mM; Merck) for 1 h before PSE/Phy were applied. The cell suspension was washed in PBS and stained using Annexin V–Alexa Fluor^®^ 647 conjugate (Thermo Scientific) for 15 min at room temperature in the dark, followed by incubation with propidium iodide (PI, Merck) and analyses by flow cytometer (BD FACSCalibur). A minimum of 1 × 10^4^ events were analyzed per analysis, and all experiments were performed in triplicate.

### 2.9. Detection of Mitochondrial Membrane Potential (MMP) Changes

Disruption of MMP after PSE 45 µg/m and Phy 60 µM treatment (3, 24, 48 or 72 h) was analyzed by flow cytometry using 0.1 μM TMRE (Molecular Probes, Eugene, OR, USA) staining. NAC/PSE or NAC/Phy experimental groups were pre-treated with N-acetyl-L-cysteine (NAC c = 2 mM; Merck) for 1 h before PSE/Phy were applied. The stained cells were incubated for 30 min at room temperature in the dark and then washed twice with PBS, resuspended and analyzed (1 × 10^4^ cells per sample). Fluorescence was detected with 585/42 (FL-2) optical filter by flow cytometer (BD FACSCalibur). All experiments were performed in triplicate.

### 2.10. Measurement of Superoxide Anions and Reactive Oxygen Species (ROS)

Flow cytometry was used to analyze intracellularly produced oxygen radicals detected by MitoSOX^TM^Red mitochondrial superoxide indicator (Thermo Fisher) or dihydrorhodamine-123 (DHR-123, Merck), which reacts with intracellular hydrogen peroxide (ROS). Jurkat cells treated with PSE 45 µg/mL and Phy 60 µM were harvested 6, 24, 48 and 72 h after exposure, washed and resuspended in PBS. NAC/PSE or NAC/Phy experimental groups were pre-treated with N-acetyl-L-cysteine (NAC c = 2 mM; Merck) for 1 h before PSE/Phy were applied. The samples were incubated for 15 min in the dark with DHR-123 (0.2 µM) and MitoSOX red (5 µM) and, after incubation, were immediately placed on ice. Fluorescence was detected with 530/30 (FL-1) and 585/42 (FL-2) optical filters, respectively, by flow cytometer (BD FACSCalibur). A minimum of 1 × 10^4^ events were analyzed per analysis, and all experiments were performed in triplicate.

### 2.11. Western Blot

Proteins isolated from Jurkat cell lysates were quantified by the Pierce^®^ BCA Protein Assay Kit (Thermo Scientific, Rockford, IL, USA) using bovine serum albumin as the standard and measured by an automated Cytation^TM^ 3 Cell Imaging Multi-Mode Reader (Biotek) at a wavelength of 570 nm. A 10% SDS-PAA gel was used to separate proteins at 100 V for 2 h. Then, proteins from the gel were transferred to a polyvinylidene difluoride (PVDF) membrane using the iBlot dry blotting system (Thermo Scientific). The membrane with the transferred proteins was incubated in 5% BSA in TBS-Tween (pH 7.4) for 1 h to minimize non-specific binding. Subsequently, the transferred membrane was incubated at 4 °C overnight with primary antibodies (Table 2). The next day, the membrane washing procedure was performed in TBS-Tween (3 × 5 min), and then the membrane was incubated with horseradish peroxidase (HRP)-conjugated anti-rabbit or anti-mouse secondary antibodies (1:1000 dilution) for 1 h at room temperature. After incubation and washing in TBS-Tween (3 × 5 min), the expression of the protein was detected using a chemiluminescent ECL substrate (Thermo Scientific) and MF-ChemiBIS 2.0 Imaging System (DNR BIO-Imaging Systems, Jerusalem, Israel). The detected band was then analyzed densitometrically using the Image Studio Lite software (LI-COR Biosciences, Lincoln, NE, USA). Equal loading was verified using antibodies against β-actin. Detection was performed in triplicate. Densitometry graphs from all Western blot analyses are provided in the Appendix A.

### 2.12. Statistical Analysis

Results are expressed as mean ± SD. One-way ANOVA followed by the Bonferroni multiple comparison test was used for statistical analyses. Differences were considered significant when *p* < 0.05. Throughout this paper, the symbols indicate the following: * *p* < 0.05, ** *p* < 0.01, *** *p* < 0.001 vs. untreated control; + *p* < 0.05, ++ *p* < 0.01, +++ *p* < 0.001 vs. PSE; and # *p* < 0.05, ## *p* < 0.01, ### *p* < 0.001 vs. Phy.

## 3. Results and Discussion

A diagnosis of acute lymphoblastic leukemia (ALL) is still associated with a poor prognosis and limited treatment options for pediatric/adult patients. Mitigation of the main side effects of conventional and new T-ALL therapies is currently a challenge. To address these issues, the use of natural compounds with prospective anti-leukemic potential and minimal side effects might be helpful. Therefore, in our work, we focused on the possible use of extract from the lichen *Pseudevernia furfuracea* (PSE) and the metabolite physodic acid (Phy) as anti-leukemic agents. Although few data are available on this topic, several studies have described that PSE or Phy has anti-proliferative, pro-apoptotic and EMT-modulating potential in various tumor cell lines [16,17,21]. In addition, recently, the screening of PSE extract in the context of cancer and inflammation was realized on a human monocytic-like cell line (THP-1) derived from a boy with acute monocytic leukemia (AML) [29]. Moreover, as seen in Figure 1A,B, we confirmed the concentration- and time-dependent cytotoxic effects and anti-proliferative potential of PSE and Phy on the Jurkat T-ALL in vitro model. From the tested concentration range (10–100), the IC50 values (used in subsequent experiments) for PSE (45 µg/mL) and Phy (60 µM) were calculated.

In addition to the anti-proliferative potential of PSE and Phy, we demonstrated that both the extract and physodic acid disrupted the mitochondrial membrane potential (MMP) (Figure 1C) in a time-dependent manner. The results show that PSE extract had a stronger effect compared to Phy. It is a fact that mitochondrial dysfunctions represent one of the possible inducers of programmed cell death, and loss of MMP may be the first sign of incipient cell death. Similar to our work, extract from the lichen *Usnea intermedia* induced alterations in MMP in breast and lung carcinoma cell lines, followed by apoptosis induction [30]. Mitochondrial damage was also described in our previous paper, where MMP in HeLa cells decreased after treatment with gyrophoric acid from the lichen *Umbilicaria hirsuta* [31]. In addition to this result, cell cycle progression and apoptosis occurrence were investigated after PSE and Phy treatment. As is known, the anti-tumor effects of lichens are mediated through the induction of cell cycle arrest at G0/G1, S or G2/M phases, followed by apoptosis, necrosis or autophagy processes [32,33,34]. In the evaluation of cell cycle progression (Table 3, Figure 1D) after PSE treatment in Jurkat cells, we observed S-phase arrest only at 24 h. The cell cycle arrest disappeared at 48 h, but the increased accumulation of cells in the sub-G0/G1 population (recognized as apoptotic from fractionated DNA) with a concomitant decrease in cells in G1 and G2/M phases of the cell cycle persisted from 24 to 72 h after PSE treatment. On the other hand, cell cycle analyses after Phy treatment showed G1 arrest at 24 and 48 h in Jurkat cells. Stronger and longer G1 arrest after Phy treatment weakened and delayed the increase in cells in the sub-G0/G1 population with a concomitant decrease in G1 and G2/M phases when compared with PSE treatment. The published data on *P. furfuracea* and physodic acid describe only a significant increase in cells in the sub-G0/G1 population, and no cell cycle arrest was detected in the human melanoma and colon cancer cell lines FemX and LS174 [35]. These findings suggest tissue-origin-specific and concentration-specific impacts of PSE and Phy on cell cycle progression.

The cell cycle analysis clearly shows that PSE and Phy induce cell death. In the next phase, we focused on the analyses of early and late events leading to the initiation and progression of programmed cell death. The Annexin V analyses (Table 4, Figure 1E), which reflected phosphatidyl serine (PS) externalization as a marker of early apoptosis [36], showed that PSE and Phy induced PS externalization and initiated cell death in Jurkat cells after 24 h of treatment. Moreover, it is clear that G1 arrest after Phy treatment delayed the initial phase of apoptosis more effectively than S-phase arrest after PSE treatment. The data show that at 72 h, only 5.3 ± 1.4% and 35.6 ± 4.6% of Jurkat cells were still alive after PSE and Phy treatment. The pro-apoptotic potential of *P. furfuracea* extract and Phy was similar, as described in a few known sources [25,26,27].

Moreover, for the first time, we demonstrated more precise mechanisms of PSE and Phy treatment in Jurkat cells. As is shown in Figure 2, the intrinsic mitochondrial form of apoptosis with caspase dependence was initiated and executed after PSE and Phy treatment. As we describe above, disrupted MMP can start mitochondria-localized processes and apoptotic signaling. In accordance with this fact, we observed the significant release of cytochrome *c* (Figure 2A) from the intermembrane space to the cytosol and caspase-9 activation (Figure 2B) shortly after 24 h of treatment. Cytochrome *c*, Apaf-1 and dATP form scaffolds that attract procaspase-9 and form the apoptosome complex with the proteolytic function needed during the apoptosis process [37]. Dimerization of procaspase-9 unleashes its catalytic domain and putative active site for caspase-3 cleavage and leads to the execution of apoptosis [38]. Subsequently, active caspase-3 cleaves protein poly (ADP-ribose) polymerase (PARP), which is responsible for several cellular processes, such as the regulation of genomic stability, DNA repair, transcription and apoptosis [39]. In the present study, time-dependent (24–72 h) caspase-3 activation (Figure 2C) and PARP cleavage (Figure 2D) were found after PSE and Phy treatment. Although these mechanisms have not been previously described after PSE and Phy treatment in Jurkat cells, similar caspase activation was confirmed in melanoma [23].

In addition, in order to determine the cytotoxic effects of PSE and Phy treatment, reactive oxygen species (ROS) were analyzed. It is known that lichen extracts and isolated/purified secondary metabolites can change the redox status of cancer cells and ROS production or accumulation, which significantly contribute to the initiation and execution of apoptosis, resulting in cytotoxicity [31,40,41]. It is also known that ROS are needed to oxidize cytochrome *c*, which is then capable of efficient apoptosome formation. The reduced cytochrome *c* might be a competitive inhibitor of Apaf-1, limiting apoptosis signaling [42]. The analyzed peroxide anion levels (referred to as ROS in general) and the superoxide anion radicals indicate the occurrence of oxidative stress after PSE and Phy treatment in Jurkat cells. As we documented, significant increases in superoxide (Figure 3A) and ROS (Figure 3B) were observed shortly after 6 h of PSE and Phy treatment with an increasing trend up to 48–72 h. Oxidative stress was more prominent with *P. furfuracea* extract than physodic acid in Jurkat cells. The pro-oxidative effects of PSE treatment were also described by Šeklič et al. [27] in an in vitro colon cancer model, where a dose- and time-dependent increase in superoxide occurred. Moreover, the authors observed glutathione level changes that mimicked oxygen species production after PSE treatment. It is a widely accepted fact that the activation of the intracellular antioxidant defense system (superoxide dismutases, catalase, glutathione) is mediated by ROS generation, and the balance between the scavenging activity of cellular antioxidant enzymes and redox status determine cell survival [43]. According to our results, Jurkat cells were under great oxidative stress after PSE and Phy treatment, which induced enhanced cytotoxicity through apoptosis. We demonstrated that ROS production affected the expression of superoxide dismutases 1 and 2 (SOD1, 2) in a time-dependent manner in Jurkat cells according to the treatment (Figure 3C). The PSE treatment decreased SOD1 and SOD2 expression after 48 and 72 h, while Phy minimally affected SOD expression. It is evident that the effects of Phy on SODs reflect ROS production, cell cycle arrest and apoptosis occurrence, which are observed at a larger scale after 72 h. In the published data, Phy treatment, as described by Sahin et al. [44], resulted in an increase in mRNA SOD levels in an in vitro hepatic cancer model when tested for antioxidant potential.

Furthermore, to elucidate whether PSE- and Phy-mediated oxidative stress is one of the main initiating factors causing cytotoxicity and apoptosis in Jurkat cells, we used ROS scavenging treatment (N-acetylcysteine; NAC). As shown in Figure 3A,B, NAC pre-treatment was able in part to scavenge reactive oxygen species after PSE and Phy treatment. Moreover, we demonstrated that ROS scavenging partially prevented MMP loss (Figure 4A) and protected mitochondria from membrane damage. In addition, reduced levels of ROS manifested in decreased apoptosis occurrence and increased survival of Jurkat cells in a time-dependent manner, as shown by Annexin V/PI analyses (Figure 4B–D).

Oxidative stress mediated by lichen secondary metabolites or extracts also represents strong genotoxic pressure accompanied by DNA damage. In our previous work, we clearly confirmed the direct association of oxygen radicals in gyrophoric acid-mediated DNA damage with apoptosis in HeLa cells [31]. Moreover, several studies have described DNA damage and genotoxicity caused by lichen secondary metabolites in vitro [24,41,45]. To analyze the impact of PSE- and Phy-mediated oxidative stress on DNA in Jurkat cells, DNA double-strand breaks and 8-oxoguanine formation were analyzed as biomarkers of DNA damage. Furthermore, the DNA repair machinery, represented by ATM (ataxia–telangiectasia mutated kinase), SMC1 protein (structural maintenance of chromosomes 1) and histone H2A.X, was activated as a result of DNA damage. These mechanisms are collectively termed the DNA damage response (DDR) and are responsible for part of the cell cycle, DNA damage detection and DNA repair [46]. As shown in Figure 5A, the levels of 8-oxo-7,8-dihydroguanine, the main product of oxidative DNA damage, increased after PSE and Phy treatment at all time periods (24–72 h). These oxidized nucleotides affect DNA strand integrity and cause DNA single- or double-strand breaks that are recognized by DDR. The subsequent regulation of downstream targets such as histone H2A.X [47], SMC1 kinase or p53 by ATM and Rad3-related (ATR) kinase phosphorylation represents an early step in the DDR machinery [48]. The presented results show that PSE- and Phy-mediated DNA damage caused double-strand breaks occurring as early as 24 h after treatment, when increased phosphorylation of ATM, as the first response element, was recognized (Figure 5B). Activated ATM kinase subsequently phosphorylated downstream factors, including histone variant H2A.X (Figure 5C) and SMC1 (Figure 5D), 24–72 h after treatment.

As we describe above, PSE and Phy treatment caused G1 or S cell cycle arrest in Jurkat cells. The recognized disruption of cell cycle progression is in accordance with oxidative stress and DNA damage mediated by PSE and Phy treatment. In addition, as a response to DNA damage, cell cycle checkpoints can be activated in the G1/S phase and at the G2/M transition [49]. The G1 checkpoint is triggered by phosphorylation and activation of checkpoint kinase 2 (Chk2) as a downstream kinase in the ATM cascade [50]. Moreover, the G1/S checkpoint is critically dependent on p53 activation by ATM. Phosphorylation of p53 by ATM stabilizes p53 and induces the expression of p21, which binds and further inhibits cyclin A/Cdk2 and cyclin E/Cdk2 complexes and the phosphorylation of pRB protein [51]. P21 is also known to inhibit the kinase activity of cyclin A/CDK1 and 2, resulting in S-phase cell cycle arrest [52]. An additional regulator of cell cycle progression in the G1/S phase is protein p27, which also inhibits phosphorylation of Rb by cyclins and therefore prevents the transcription of genes required for the G1/S transition [53]. There is evidence supporting the fact that lichen secondary metabolites can also affect these checkpoint cascades. It was reported that usnic acid [54], ramalin [55] and extract of *Parmotrema reticulatum* [56] inhibited cell proliferation via cell cycle arrest, increased the expression of the CDK inhibitor p21/cip1 protein and upregulated TP53. According to these facts, we analyzed changes in several cell cycle checkpoint proteins. As the results show, PSE- and Phy-mediated DNA damage also activated the expression of cell cycle checkpoint protein p53 by ATM (Figure 6A). Moreover, we confirmed that G1/S checkpoint activation led to a downstream cascade resulting in cell cycle arrest and apoptosis. For the first time, we also demonstrated that ATM and p53 phosphorylation in response to PSE- and Phy-mediated DNA damage activated the expression of p21 (Figure 6B) and p27 kinases (Figure 6C), which subsequently inhibited the phosphorylation of Rb protein (Figure 6C).

An important mechanism in apoptosis control is mitogen-activated protein kinase (MAPK) regulation [57]. It is known that oxidative stress-induced modulation of stress/survival/apoptotic pathways includes p38 MAPK, extracellular signal-regulated kinases (Erk 1/2), c-Jun N-terminal kinase (JNK) and PI3K/Akt. Generally, increased phosphorylation of these kinases led to the promotion of cell death in a time-dependent manner. Moreover, the existence of crosstalk between Akt and ATM and Akt-mediated regulation of cell cycle proteins such as p53 and p27 impact cell fate decisions in stress conditions [58,59]. To date, no data have been published about the modulation of these pathways by PSE- or Phy-induced apoptosis in Jurkat cells. On the other hand, the contribution of lichens to the regulation of p38 MAPK, JNK and PI3K/Akt in apoptosis signaling was described [34,60,61,62]. To study the effect of PSE and Phy treatment on stress/survival pathway protein activation, we analyzed the phosphorylation status of the above-mentioned proteins in Jurkat cells (Figure 7). We observed that PSE treatment significantly increased the phosphorylation of JNK (24 h), p38 MAPK (24–48 h) and Akt (24–72 h) with upstream phosphorylation of PI3K kinase (3–24 h) in a time-dependent manner. Furthermore, Phy treatment increased the phosphorylation of JNK (24 h), p38 MAPK (48 h), PI3K (3–24 h) and Akt (72 h). The pro-apoptotic phosphorylation of p38 MAPK (24 or 48 h) and Akt (72 h) culminated differently between PSE and Phy treatment due to delayed apoptosis and cell cycle arrest in the case of Phy. Moreover, increased Akt phosphorylation is in accordance with the increased activation of ATM (Figure 5; DNA damage response) and inhibition of p53, Rb and p27 cell cycle checkpoint proteins (Figure 6). In addition, increased phosphorylation of JNK (24 h) and p38 MAPK (48 h) clearly reacted to oxidative stress induced by PSE and Phy treatment, where we noticed elevated levels of ROS and superoxide (Figure 3). The total protein levels were mostly unaffected at early time points, but at 48 and 72 h, degradation resulting from the execution of apoptosis was observed in response to PSE or Phy treatment. Taken together, the activation of the above-mentioned kinases corresponds to the pro-apoptotic effect of PSE and Phy treatment and is mediated by oxidative stress, DNA damage and cell cycle arrest, to which MAPK kinases reacted.

Finally, safety and minimal side effects on healthy tissues are important indicators during the discovery of new drugs in cancer research. Therefore, we additionally analyzed normal T cells in our experiments. As is shown in Figure 8, we observed that PSE and Phy treatment had no or minimal, non-significant cytotoxic effects on survival, MMP changes, cell cycle progression and PS externalization or apoptosis occurrence in peripheral T lymphocytes. Similarly, Emsen et al. [63] showed that physodic acid isolated from *P. furfuracea* only caused cytotoxicity in cultured human lymphocytes when it was administered at high doses. 

In conclusion, our results show that PSE and Phy treatment induced apoptosis in Jurkat cells in a dose- and time-dependent manner. We confirmed that ROS production and consequent DNA damage played an eminent role in PSE- and Phy-mediated apoptosis. Moreover, the DNA repair mechanism, including the phosphorylation of ATM, HA2.X and SMC1 proteins, was subsequently activated, followed by p21, p53 and p27 activation and cell cycle arrest. Moreover, PSE and Phy treatment led to the phosphorylation of MAPK signaling, including p38 MAPK, JNK and PI3K/Akt. Furthermore, minimal or no cytotoxicity in normal peripheral lymphocytes supports the use of PSE and Phy as anti-leukemic agents. We assume that the stronger effects of the PSE extract compared to the physodic acid metabolite on the monitored parameters are due to the synergistic or additive effects and interactions of the individual components of the extract.

## Figures and Tables

**Figure 1 pharmaceutics-13-02173-f001:**
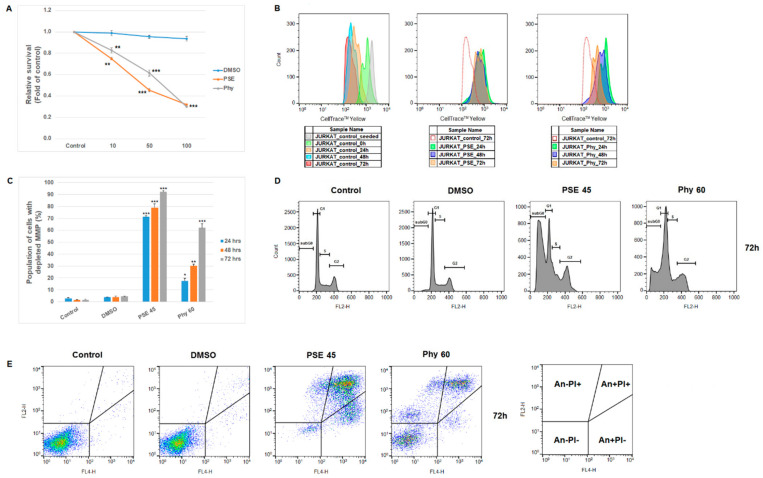
Effect of PSE and Phy on proliferation, apoptosis, cell cycle and MMP changes in Jurkat cells. (**A**) Relative survival analyses of Jurkat cells treated with PSE, Phy and DMSO concentration range at 72 h. (**B**) Flow cytometric analyses of cell proliferation at 24–72 h after PSE and Phy treatment. (**C**) Analyses of mitochondrial membrane potential changes after PSE and Phy treatment. (**D**) Representative diagrams of cell cycle phases (G1, S, G2) analyses after 72 h of PSE and Phy treatment. (**E**) Representative dot plots of Annexin V/PI apoptosis occurrence after 72 h of PSE and Phy treatment. The presented data are from three independent experiments. Significantly different at * *p* < 0.05, ** *p* < 0.01, *** *p* < 0.001 vs. untreated cells (control).

**Figure 2 pharmaceutics-13-02173-f002:**
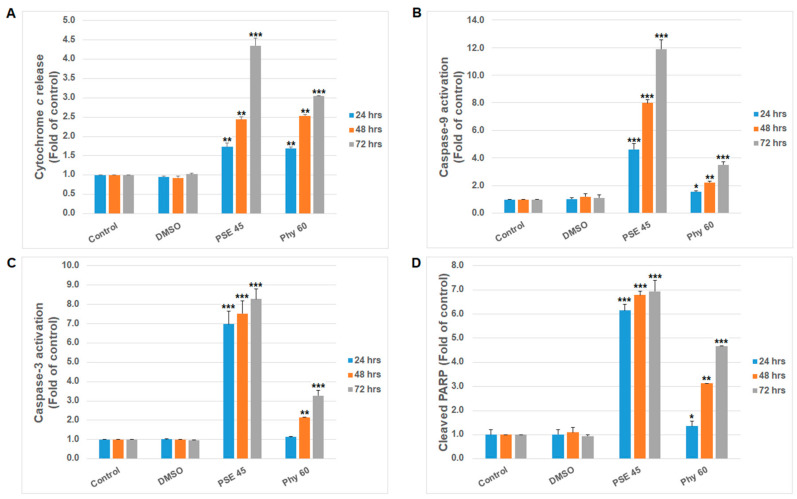
Intrinsic apoptosis pathway analyses after PSE and Phy treatment in Jurkat cells. (**A**) Caspase-9 activation, (**B**) cytochrome *c* release, (**C**) caspase-3 activation and (**D**) PARP cleavage at 24–72 h after PSE (45 µg/mL) and Phy (60 µM) treatment in Jurkat cells. The presented data are from three independent experiments. Significantly different at * *p* < 0.05, ** *p* < 0.01, *** *p* < 0.001 vs. untreated cells (control).

**Figure 3 pharmaceutics-13-02173-f003:**
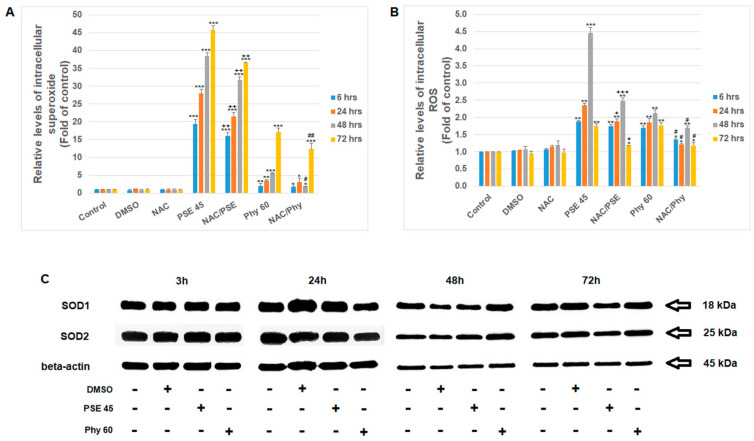
Oxidative stress analyses in Jurkat cells after PSE and Phy treatment. (**A**) Flow cytometry analyses of intracellular superoxide anion content, (**B**) ROS generation and (**C**) Western blot analyses of SOD1/2 expression profile after PSE (45 µg/mL) and Phy (60 µM) treatment. The presented data are from three independent experiments. Significantly different at * *p* < 0.05, ** *p* < 0.01, *** *p* < 0.001 vs. untreated cells (control); + *p* < 0.05, ++ *p* < 0.01, +++ *p* < 0.001 vs. PSE; and # *p* < 0.05, ## *p* < 0.01 vs. Phy.

**Figure 4 pharmaceutics-13-02173-f004:**
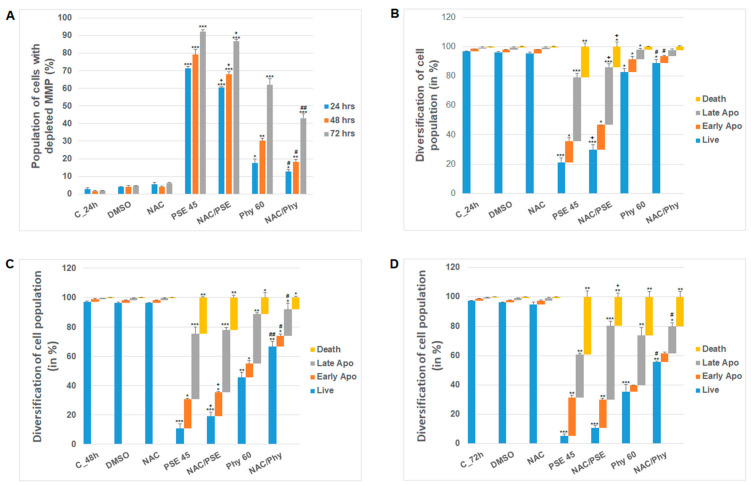
NAC pre-treatment effect on (**A**) MMP and (**B**–**D**) apoptosis occurrence in Jurkat cells after PSE (45 µg/mL) and Phy (60 µM) treatment. The presented data are from three independent experiments. Significantly different at * *p* < 0.05, ** *p* < 0.01, *** *p* < 0.001 vs. untreated cells (control); + *p* < 0.05 vs. PSE; and # *p* < 0.05, ## *p* < 0.01 vs. Phy.

**Figure 5 pharmaceutics-13-02173-f005:**
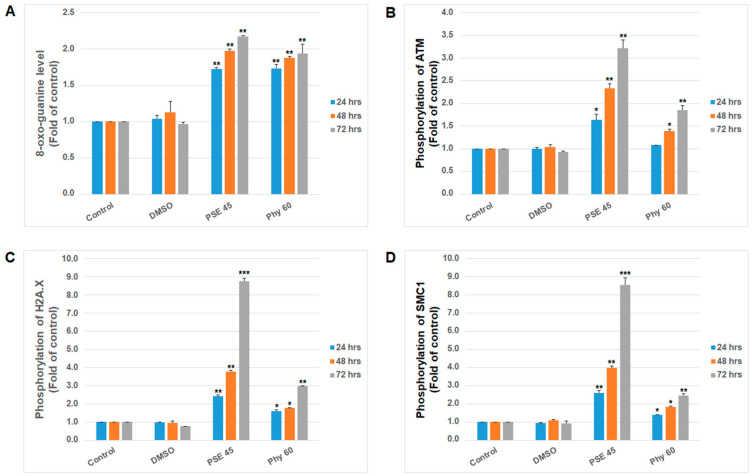
Effect of PSE (45 µg/mL) and Phy (60 µM) treatment on (**A**) 8-oxoguanine levels and phosphorylation status of (**B**) ATM kinase, (**C**) histone H2A.X and (**D**) SMC1 kinase in Jurkat cells analyzed by flow cytometry. The presented data are from three independent experiments. Significantly different at * *p* < 0.05, ** *p* < 0.01, *** *p* < 0.001 vs. untreated cells (control).

**Figure 6 pharmaceutics-13-02173-f006:**
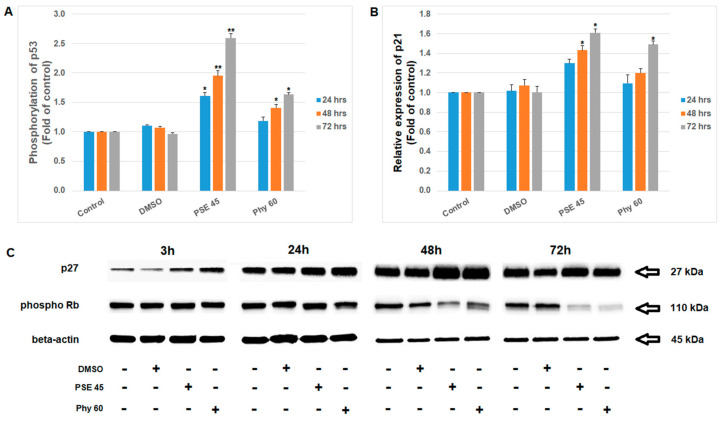
Effect of PSE (45 µg/mL) and Phy (60 µM) treatment on (**A**) p53 phosphorylation, (**B**) p21, (**C**) p27 expression and Rb phosphorylation in Jurkat cells detected by flow cytometry and Western blot. The presented data are from three independent experiments. Significantly different at * *p* < 0.05, ** *p* < 0.01 vs. untreated cells (control).

**Figure 7 pharmaceutics-13-02173-f007:**
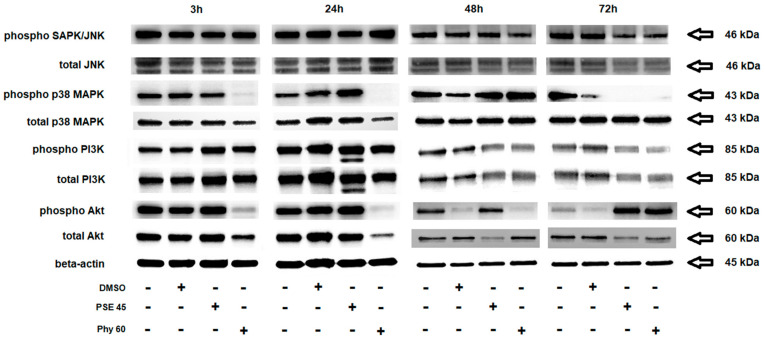
Western blot analyses of stress/survival/apoptotic signaling pathways. Effect of PSE (45 µg/mL) and Phy (60 µM) treatment on total levels and phosphorylation of JNK, p38 MAPK, PI3K and Akt proteins in Jurkat cells. The presented data are from three independent experiments.

**Figure 8 pharmaceutics-13-02173-f008:**
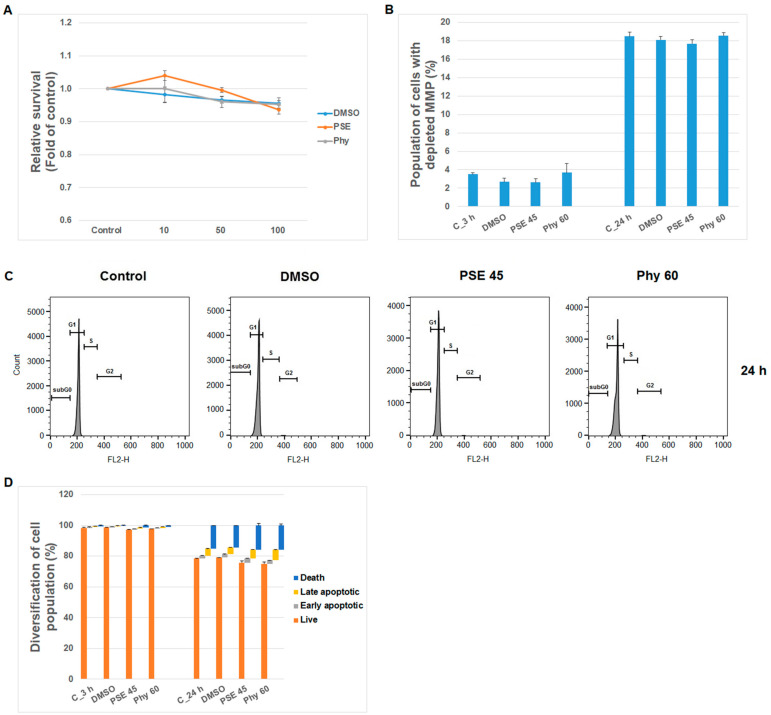
Human T-lymphocyte response after PSE and Phy treatment. Effect of PSE (45 µg/mL) and Phy (60 µM) treatment (3–24 h) on (**A**) cell survival, (**B**) MMP changes, (**C**) cell cycle phases (G1, S, G2) progression and (**D**) apoptosis occurrence in HTLs.

**Table 1 pharmaceutics-13-02173-t001:** Flow cytometry antibodies and dye list used for staining.

Primary Antibodies	
Cleaved Caspase-3 (Asp175) (5A1A) Rabbit mAb PE conjugate	Cell Signaling Technology^®^
Cleaved Caspase-9 (Asp315) (D8I9E) Rabbit mAb PE conjugate	
Cleaved PARP (Asp214) (D64E10) XP Rabbit mAb PE conjugate	
Phospho-Histone H2A.X (Ser139) (20E3) Rabbit mAb Alexa Fluor 647 conjugate	
Phospho-p53 (Ser15) (16G8) Mouse mAb PE conjugate	
p21 Waf1/Cip1 (12D1) Rabbit mAb	
Anti-phospho-ATM mAb PE conjugate	Merck Millipore
Anti-phospho-SMC1 mAb Alexa Fluor^®^ 488 conjugate	
Cytochrome *c* (6H2) Mouse mAb FITC conjugate	Thermo Scientific
Anti-Oxoguanine 8 (2Q2311) Mouse mAb	Abcam
**Secondary Conjugated Antibodies**	
Goat anti-Mouse IgG Alexa Fluor^®^ 488 conjugate	Thermo Scientific
Goat anti-Rabbit IgG Alexa Fluor^®^ 488 conjugate	

**Table 2 pharmaceutics-13-02173-t002:** Antibodies used for immunoblotting.

Primary Antibodies	Mr (kDa)	Source/Origin	Company
JNK1	46	Mouse	Thermo Scientific
Phospho SAPK/JNK (Thr183/Tyr185)	46	Mouse	Cell Signaling Technology^®^
Phospho Rb (Ser807/811)	110	Rabbit	
Phospho p38 MAPK (Thr180/Tyr182)	43	Rabbit	
p38 MAPK	43	Rabbit	
Phospho Akt (Thr308)	60	Rabbit	
Akt (pan)	60	Rabbit	
Phospho PI3K p85 (Tyr458)/p55 (Tyr199)	85, 60	Rabbit	
PI3K p85	85	Rabbit	
p27 Kip1	27	Rabbit	
SOD1/CuZnSOD	18	Rabbit	
β-actin	45	Mouse	
SOD2/MnSOD	25	Mouse	Abcam
**Secondary Antibodies**	**Mr (kDa)**	**Source/Origin**	**Company**
Anti-Mouse IgG HRP	-	Goat	Cell Signaling Technology^®^
Anti-Rabbit IgG HRP	-	Goat	

**Table 3 pharmaceutics-13-02173-t003:** Cell cycle analysis of Jurkat cells after 24, 48 and 72 h incubation with PSE and Phy.

Jurkat	Live (An^−^/PI^−^)	Early Apo (An^+^/PI^−^)	Late Apo (An^+^/PI^+^)	Death (An^−^/PI^+^)
CTRL 24 h	96.8 ± 0.3	1.8 ± 0.1	1.0 ± 0.3	0.5 ± 0.1
DMSO	96.2 ± 0.6	1.8 ± 0.2	1.5 ± 0.3	0.5 ± 0.2
PSE 45	21.2 ± 2.9 ***	14.5 ± 2.1 *	43.5 ± 2.6 ***	20.9 ± 2.4 **
Phy 60	82.7 ± 2.4 *	8.8 ± 1.9 *	6.5 ± 0.8 *	2.0 ± 0.2
CTRL 48 h	97.0 ± 0.6	1.8 ± 0.6	0.6 ± 0.1	0.5 ± 0.1
DMSO	96.2 ± 0.6	1.8 ± 0.4	1.5 ± 0.2	0.5 ± 0.1
PSE 45	10.7 ± 3.2 ***	19.9 ± 0.9 *	44.8 ± 4.5 ***	24.7 ± 0.4 **
Phy 60	45.6 ± 3.6 **	9.4 ± 2.0 *	33.7 ± 1.0 **	11.4 ± 3.5 *
CTRL 72 h	97.3 ± 0.4	1.4 ± 0.2	0.8 ± 0.2	0.4 ± 0.1
DMSO	96.3 ± 0.2	1.5 ± 0.1	1.5 ± 0.1	0.8 ± 0.2
PSE 45	5.3 ± 1.4 ***	26.3 ± 1.7 **	29.2 ± 0.9 **	39.3 ± 4.0 **
Phy 60	35.6 ± 4.6 ***	4.2 ± 0.1	34.2 ± 1.1 **	26.0 ± 2.6 **

The presented data are from three independent experiments after 24, 48 and 72 h treatment as average percentage ± SD. Significantly different at * *p* < 0.05, ** *p* < 0.01, *** *p* < 0.001 vs. untreated cells (control).

**Table 4 pharmaceutics-13-02173-t004:** Apoptosis analysis of Jurkat cells after 24, 48 and 72 h incubation with PSE and Phy.

Jurkat	Live (An^−^/PI^−^)	Early Apo (An^+^/PI^−^)	Late Apo (An^+^/PI^+^)	Death (An^−^/PI^+^)
CTRL 24 h	96.8 ± 0.3	1.8 ± 0.1	1.0 ± 0.3	0.5 ± 0.1
DMSO	96.2 ± 0.6	1.8 ± 0.2	1.5 ± 0.3	0.5 ± 0.2
PSE 45	21.2 ± 2.9 ***	14.5 ± 2.1 *	43.5 ± 2.6 ***	20.9 ± 2.4 **
Phy 60	82.7 ± 2.4 *	8.8 ± 1.9 *	6.5 ± 0.8 *	2.0 ± 0.2
CTRL 48 h	97.0 ± 0.6	1.8 ± 0.6	0.6 ± 0.1	0.5 ± 0.1
DMSO	96.2 ± 0.6	1.8 ± 0.4	1.5 ± 0.2	0.5 ± 0.1
PSE 45	10.7 ± 3.2 ***	19.9 ± 0.9 *	44.8 ± 4.5 ***	24.7 ± 0.4 **
Phy 60	45.6 ± 3.6 **	9.4 ± 2.0 *	33.7 ± 1.0 **	11.4 ± 3.5 *
CTRL 72 h	97.3 ± 0.4	1.4 ± 0.2	0.8 ± 0.2	0.4 ± 0.1
DMSO	96.3 ± 0.2	1.5 ± 0.1	1.5 ± 0.1	0.8 ± 0.2
PSE 45	5.3 ± 1.4 ***	26.3 ± 1.7 **	29.2 ± 0.9 **	39.3 ± 4.0 **
Phy 60	35.6 ± 4.6 ***	4.2 ± 0.1	34.2 ± 1.1 **	26.0 ± 2.6 **

The presented data are from three independent experiments after 24, 48 and 72 h treatment as average percentage ± SD. Significantly different at * *p* < 0.05, ** *p* < 0.01, *** *p* < 0.001 vs. untreated cells (control).

## Data Availability

The data presented in this study are available in the Appendix A or can be provided by the authors.

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
