# Peer review of "Pro-Apoptotic Potential of *Pseudevernia furfuracea* (L.) *Zopf* Extract and Isolated Physodic Acid in Acute Lymphoblastic Leukemia Model In Vitro"

_pharmaceutics, 2021, doi:10.3390/pharmaceutics13122173_

Round 1
Reviewer 1 Report
Dear Authors,
Please find below my comments on this manuscript.
This work sets out to characterize the Acute lymphoblastic leukaemia of Pseudevernia furfuracea secondary metabolite physodic acid. In general. I feel that this manuscript adds valuable information to anti-cancer potential of physodic acid. It contains interesting data and results. However, there are certain issues which should be addressed prior to the acceptance of the manuscript.
I have listed these concerns below and I am sure addressing them will broaden the scientific appeal of the manuscript.
Thank you very much.
Best regards.
Major comments
- Where does physodic acid stand in comparison to the other lichen compounds showing similar activity? A comparison of activity is essential to place this paper in the bigger picture of natural product research. You did mention the other compounds with similar activity. A comparative table or text on this might be helpful to guide the potential readers.
- In most of the cases natural products are not suitable to be used as such as medicine, instead natural product analogs are. An essential component of natural product implementation is drug industry therefore is the information on the genes involved in the synthesis of the natural product. Is anything known about the genes that might be synthesizing physodic acid? It is important to state how the combined knowledge can be mended for the pharmaceutical usage.
- The language needs a thorough revision. After a certain point the grammatical errors were too many to list, so I decided to skip that part altogether and just focus on the scientific quality of the manuscript.
Minor comments
- Abstract Line 18, Please change: “these naturally occurred lichen secondary” to “these naturally occurring lichen secondary….”
- Please add citations to “Especially during the induction phase of ALL therapy, the serious toxicities were ob- 38 served leading to neutropenia and infection.
- Lines 39-40. Sentence “The survival rates for ALL are contributed 39 with a key component of chemotherapy, the vincristine” is grammatically incorrect. Please rephrase
- Therefore, identification of novel natural or synthetic substances with reduced toxicity and retained anti-cancer activity in ALL treatment is challenging approach in research area [3-8].
Do you mean “is the major challenge”?
8. Recently, pharmaceutical potential of lichens and their secondary metabolites have increased attention worldwide as reviewed.
Citations required
9. Backorova 53 et al. [12] found that atranorin inhibits cell proliferation, induce apoptosis and initiate ac- 54 cumulation of leukaemia cells in S-phase of cell cycle
Please change to: “Backorova 53 et al. [12] found that atranorin inhibits cell proliferation, induces apoptosis and initiates ac- 54 cumulation of leukaemia cells in S-phase of cell cycle”
10. Lines 68-69 Please rephrase this sentence: In our best knowledge, no data exist in these intentions.
- In the discussion lines 231-238 represent knowledge gap which is usually the part of the introduction. Shorten this to one line and move the rest to the introduction, making sure that this information is not repeated in the introduction
- Lines 242-244. This statement is not entirely correct. A recent study by Ingelfinger et al (Frontiers in pharmacology, 2020) tested the bioactivity of Pseudevernia furfuracea extract, including anti-proliferative property, on cancer cells. Of course, it could be the olivetoric acid chemotype pf Pseudevenia furfuracea but the possibility that it is physodic cannot be excluded so it is worth mention that study and their results. Please relate your study to this and other recent studies if any.
Author Response
Reviewer 1
Please find below my comments on this manuscript.
This work sets out to characterize the Acute lymphoblastic leukaemia of Pseudevernia furfuracea secondary metabolite physodic acid. In general. I feel that this manuscript adds valuable information to anti-cancer potential of physodic acid. It contains interesting data and results. However, there are certain issues which should be addressed prior to the acceptance of the manuscript.
I have listed these concerns below and I am sure addressing them will broaden the scientific appeal of the manuscript.
Thank you very much.
Best regards.
We thank reviewer for her/his valuable effort to improve manuscript.
Major comments
- Where does physodic acid stand in comparison to the other lichen compounds showing similar activity? A comparison of activity is essential to place this paper in the bigger picture of natural product research. You did mention the other compounds with similar activity. A comparative table or text on this might be helpful to guide the potential readers.
Thank you for the opinion.
The published data indicate that different lichen metabolites or extracts acts through different signalling pathways in various cell types. For example, usnic acid does not consistently induce cell cycle blocks in all cell types. Also cell survival, proliferation and induction of cell death is concentration-, time- and cell type-specific. In our project we focused on very unexplored area of anti-leukemic potential of lichen extract and physodic acid. As we added to Introduction section, mostly only screening of cytotoxic potential of some extract and substances from lichens were tested by metabolic and proliferative tests (MTT…) in leukemia cell models. Now we added this statement to Introduction. Only 2 papers mentioned physodic acid (Talapatra 2016) and PSE (Ingelfinger 2020) specifically in relation to leukemia as we citated in Introduction or Results and discussion section.
In the Results and discussions section, we compared our particular findings with several other metabolites (gyrophoric acid, usnic acid, atranorin, ramalin) or extracts from different origin or in different tumor cell models.
A comparison of activity of different metabolites and extracts with PSE extract and physodic acids is good idea but for nice robust cancer review as we preparing now as a chapter to the book.
2. In most of the cases, natural products are not suitable to be used as such as medicine, instead natural product analogs are. An essential component of natural product implementation is drug industry therefore is the information on the genes involved in the synthesis of the natural product. Is anything known about the genes that might be synthesizing physodic acid? It is important to state how the combined knowledge can be mended for the pharmaceutical usage.
Thank you for the opinion.
Based on our knowledge there is no information on genes involved in synthesis of physodic acid. We searched in two main databases KEGG (Kyoto Encyclopedia of Genes and Genomes) which deals with genes and MIBIG (Minimum information about a Biosynthetic Gene cluster) which deals with gene cluster. Genes are always linked with living forms, so we have to look on organism from which was physodic acid isolated. In our case it is P. furfuracea. We searched also by lichen but nothing about synthesis of physodic acid was found.
MIBIG: https://mibig.secondarymetabolites.org/query
KEGG: https://www.genome.jp/kegg/
It is well known and published that most lichen compounds are synthesised by the primary fungal partner in the symbiosis. The main biosynthetic pathways leading to lichen compounds are the shikimic acid pathway (pulvinic acid derivatives), the mevalonic acid pathway (terpenes, carotenoids), and the polyketide pathway (depsides, depsidones, depsones, usnic acids, anthraquinones, xanthones) (Elix and Stocker-Wörgötter, 2008).
At least composition of PSE extract related to these chemical groups was added to Introduction text.
3. The language needs a thorough revision. After a certain point the grammatical errors were too many to list, so I decided to skip that part altogether and just focus on the scientific quality of the manuscript.
Language correction was made.
Minor comments
4. Abstract Line 18, Please change: “these naturally occurred lichen secondary” to “these naturally occurring lichen secondary….”
Corrected
5. Please add citations to “Especially during the induction phase of ALL therapy, the serious toxicities were ob- 38 served leading to neutropenia and infection.
Citations added according to your advice.
6. Lines 39-40. Sentence “The survival rates for ALL are contributed 39 with a key component of chemotherapy, the vincristine” is grammatically incorrect. Please rephrase
Corrected
7. Therefore, identification of novel natural or synthetic substances with reduced toxicity and retained anti-cancer activity in ALL treatment is challenging approach in research area [3-8].
Do you mean “is the major challenge”?
Thank you, we agree with your change in text. Corrected
8. Recently, pharmaceutical potential of lichens and their secondary metabolites have increased attention worldwide as reviewed.
Citations required
Thank you for your advice, we added proper citation there.
9. Backorova 53 et al. [12] found that atranorin inhibits cell proliferation, induce apoptosis and initiate ac- 54 cumulation of leukaemia cells in S-phase of cell cycle
Please change to: “Backorova 53 et al. [12] found that atranorin inhibits cell proliferation, induces apoptosis and initiates ac- 54 cumulation of leukaemia cells in S-phase of cell cycle”
Corrected
10. Lines 68-69 Please rephrase this sentence: In our best knowledge, no data exist in these intentions.
Corrected
11. In the discussion lines 231-238 represent knowledge gap which is usually the part of the introduction. Shorten this to one line and move the rest to the introduction, making sure that this information is not repeated in the introduction
Thank you for your valuable opinion. We moved part of the text to Introduction and deleted some duplicities.
12. Lines 242-244. This statement is not entirely correct. A recent study by Ingelfinger et al (Frontiers in pharmacology, 2020) tested the bioactivity of Pseudevernia furfuracea extract, including anti-proliferative property, on cancer cells. Of course, it could be the olivetoric acid chemotype pf Pseudevenia furfuracea but the possibility that it is physodic cannot be excluded so it is worth mention that study and their results. Please relate your study to this and other recent studies if any.
Thank you for this recommendation. We added relevant mentioned information to this section:
Besides that, recently the screening of PSE extract in the context of cancer and inflammation was realised on the human monocytic-like cell line (THP-1) derived from acute monocytic leukemia (AML) suffered boy {Ingelfinger, 2020}.
Reviewer 2 Report
Review on the Manuscript for MDPI Pharmaceutics
Pro-apoptotic potential of Pseudevernia furfuracea (L.) Zopf extract and isolated Physodic acids in Acute lymphoblastic leukaemia model in vitro.
Martin Kello1*, Tomas Kuruc1, Klaudia Petrova1, Michal Goga 2, Zuzana Michalova1, Matus Coma1, Dajana Rucova2 5 and Jan Mojzis1* 6
The paper is about a very important topic, the treatment of acute lymphoblastic leukemia (ALL) by potential use of a lichen and a secondary metabolic product of it.
The paper is generally well written and richly illustrated. Some details to check and correct are indicated directly in the text of the manuscript.
Physodic acid in the title and several places in the text should be used in the singular form.
Since no auctor names are added to the lichen names mentioned in the text, a reference (to IndexFungorum or Mycobank) should be added in Material and Methods, indicating that the nomenclature of lichen names follows these databases.
The illustrations are very small. The gray text should be changed to black since it is not easy to read.
The paper is suggested for acceptance after these minor changes.

Author Response
Reviewer 2
Pro-apoptotic potential of Pseudevernia furfuracea (L.) Zopf extract and isolated Physodic acids in Acute lymphoblastic leukaemia model in vitro.
Martin Kello1*, Tomas Kuruc1, Klaudia Petrova1, Michal Goga 2, Zuzana Michalova1, Matus Coma1, Dajana Rucova2 5 and Jan Mojzis1* 6
The paper is about a very important topic, the treatment of acute lymphoblastic leukemia (ALL) by potential use of a lichen and a secondary metabolic product of it.
The paper is generally well written and richly illustrated. Some details to check and correct are indicated directly in the text of the manuscript.
We thank reviewer for her/his valuable effort to improve manuscript.
Physodic acid in the title and several places in the text should be used in the singular form.
Corrected
Since no author names are added to the lichen names mentioned in the text, a reference (to IndexFungorum or Mycobank) should be added in Material and Methods, indicating that the nomenclature of lichen names follows these databases.
As is described in Materials section the lichen was collected, identified and characterised by botanists and lichenologists prof. Backor and Dr. Goga from Department of Botany, Institute of Biology and Ecology, Faculty of Science, Pavol Jozef Šafárik University. At our University we are working with this lichen several years. Lichen was determined by morphology as well as by chemistry. Chemical composition indicate that it is chemotype 1: containing atranorin and physodic acid (Stenross, 2016). TLC, HPLC and NMR were used for identification of substances (Petrova, 2021). Indexfungorum or Mycobank is very helpful with microscopic lichens, where determination by ITS DNA sequence data is necessary.
The illustrations are very small. The gray text should be changed to black since it is not easy to read.
Thank you for advice, we provided for production staff high dpi picture of every figures in manuscript. The colour of text was changed to black bold.
The paper is suggested for acceptance after these minor changes.
Reviewer 3 Report
In this manuscript the authors evaluate ”Pro-apoptotic potential of Pseudevernia furfuracea (L.) Zopf ex- 2 tract and isolated Physodic acids in Acute lymphoblastic leu- 3 kaemia model in vitro” The topic is interesting but I have some criticisms:
- why are the IC50s of PSE and Phy expressed in different units of measurement?
- in all experiments performed in the manuscript, the effects of PSE are always greater than Phy. Authors need to fully explain this fundamental point. Also because if PSE has a greater effect than Phy, probably no further investigations on Phy are necessary but it would be more appropriate to evaluate other metabolites of PSE.
- the results obtained in the western blots of JNK, p38 MAPK and Akt should be better described. In their present form they appear confused
Author Response
Reviewer 3
In this manuscript the authors evaluate ”Pro-apoptotic potential of Pseudevernia furfuracea (L.) Zopf ex- 2 tract and isolated Physodic acids in Acute lymphoblastic leu- 3 kaemia model in vitro” The topic is interesting but I have some criticisms:
- why are the IC50s of PSE and Phy expressed in different units of measurement?
Thank you for your question.
There must be different units due the fact that extract from Pseudevernia is composed from several different metabolites and substances. Therefore, PSE must be calculated in mass per volume. Physodic acid was calculated as chemical unique substance where more precise calculation of concentration is molarity. This fact is also important for animal studies.
- in all experiments performed in the manuscript, the effects of PSE are always greater than Phy. Authors need to fully explain this fundamental point. Also because if PSE has a greater effect than Phy, probably no further investigations on Phy are necessary but it would be more appropriate to evaluate other metabolites of PSE.
We not agree with your opinion. Necessary of Phy investigation and others secondary metabolites included in Pseudevernia extract is very important. Lichen grows very slowly and therefore sufficient dry matter for the extraction and isolation of secondary metabolites is very limiting. On the other hand, the eventual synthesis of individual metabolites as chemically pure substances is easy and in sufficient quantities. Therefore, understandings and investigation of individual substances from plant extracts (especially lichens) for their antiproliferative, proapoptotic effects in cancer therapy is important. Also as you pointed the PSE extract effects in this case were greater than physodic acid itself. But this is not a general feature if extract have stronger effect as particular substances. In extract as a mixture of several substances we can expect synergic or additive or antagonistic or blocking effects between all parts and final effect of extract is a summary of these interactions and reactions. Also it is not new that individual substances from extract have different strength of same effect or have some completely different effects. In our case, the composition of the most abundant metabolites in P. furfuracea extract (PSE) is atranorin, chloroatranorine (depside group); physodic acid, oxyphysodic acid and O-methylphysodic acid (depsidone group). Atranorin/chloroatranorin (also usnic acid) are in general most studies lichen secondary metabolites obtained in several lichen species. Therefore we studied physodic acids.
In the manuscript in introduction and conclusion I made sentence pointing to composition of extract:
Introduction: The composition of the most abundant metabolites in P. furfuracea extract (PSE) is atranorin, chloroatranorine (depside group); physodic acid, oxyphysodic acid and O-methylphysodic acid (depsidone group) [22], which together account for their overall antitumor effects through their interactions (synergy, additivity, antagonism, inhibition).
Conclusion: We assume that the stronger effects of the PSE extract compared to the physodic acid metabolite in the monitored parameters are due to the synergistic or additive effects and interactions of the individual components of the extract.
- the results obtained in the western blots of JNK, p38 MAPK and Akt should be better described. In their present form they appear confused.
We partially agree. We provided more description of results in accordance with other results and their interconnections to make bigger picture of all data.
Reviewer 4 Report
Comments:
In this manuscript, the authors described “Pro-apoptotic potential of Pseudevernia furfuracea (L.) Zopf extract and isolated Physodic acids in Acute lymphoblastic leukaemia model in vitro.”. This study shows the antiproliferative effect of P. furfuracea extract and physodic acid induced apoptosis and modulated the signalling pathways in acute lymphoblastic leukaemia model in vitro. However, there are a few points that need to be clarified.
Comment
- Anti-leukemic potential of PSE and Phy in ALL model in vitro. In this paper the author uses the human T lymphoblastic leukaemia cell line Jurkat. What is the anti-leukemia potential of PSE and Phy in other leukemia cell lines?
- In Figure 1, the graphics resolution is too poor.
- In Figure 3C, the Western blot should be quantitative. In addition, how about the expression of other antioxidant enzymes such as catalase and GPx?
- In Figures 6C and 7, the Western blot should be quantitative.
- The HPLC profile of Pseudevernia furfuracea should be included in this article.
Author Response
Reviewer 4
In this manuscript, the authors described “Pro-apoptotic potential of Pseudevernia furfuracea (L.) Zopf extract and isolated Physodic acids in Acute lymphoblastic leukaemia model in vitro.”. This study shows the antiproliferative effect of P. furfuracea extract and physodic acid induced apoptosis and modulated the signalling pathways in acute lymphoblastic leukaemia model in vitro. However, there are a few points that need to be clarified.
We thank reviewer for her/his valuable effort to improve manuscript.
Comment
- Anti-leukemic potential of PSE and Phy in ALL model in vitro. In this paper the author uses the human T lymphoblastic leukaemia cell line Jurkat. What is the anti-leukemia potential of PSE and Phy in other leukemia cell lines?
In general very little is known about anti-leukemic potential of Lichen extracts or metabolites. From other types of leukaemia’s the promyelocytic leukemia (HL-60), chronic myelogenous leukaemia (K-562) and murine leukemia (L1210) cell lines were screening only for potential antiproliferative activity (MTT, BrdU assay) with Cladonia salzmannii ether extract, atranorin, usnic acid and metabolites from genus Parmelia (reviewed in Mohammadi 2020). Only physodic acid and not PSE was tested by screening method on chronic myelogenous leukemia (K-562) cell line showing cytotoxic potential (Talapatra et al., 2016). Besides that, recently the screening of PSE extract in the context of cancer and inflammation was realised on the human monocytic-like cell line (THP-1) derived from acute monocytic leukemia (AML) suffered boy (Ingelfinger, 2020).
Mohammadi, M., Zambare, V., Malek, L., Gottardo, C., Suntres, Z., & Christopher, L. (2020). Lichenochemicals: extraction, purification, characterization, and application as potential anticancer agents. Expert opinion on drug discovery, 15(5), 575-601.
Talapatra, S. K., Rath, O., Clayton, E., Tomasi, S., & Kozielski, F. (2016). Depsidones from lichens as natural product inhibitors of M-phase phosphoprotein 1, a human kinesin required for cytokinesis. Journal of natural products, 79(6), 1576-1585.
Ingelfinger, R., Henke, M., Roser, L., Ulshöfer, T., Calchera, A., Singh, G., ... & Schiffmann, S. (2020). Unraveling the pharmacological potential of lichen extracts in the context of cancer and inflammation with a broad screening approach. Frontiers in pharmacology, 11, 1322.
The information about leukemic cell lines used in experiments with lichen extract and metabolites was added to Introduction section.
2. In Figure 1, the graphics resolution is too poor.
Thank you for advice, we provided for production staff high dpi picture of every figures in manuscript. The colour of text was changed to black bold.
3. In Figure 3C, the Western blot should be quantitative. In addition, how about the expression of other antioxidant enzymes such as catalase and GPx?
Thank you for your advice, the quantification graphs from densitometry are provided and will be published in Supplementary file. As you can see, the manuscript is robust with several methods and results. We chose only SOD 1 and 2 as most common antioxidant enzymes. We know that several others antioxidants system could be involved but the list of these enzymes is long and we didn't have a chance to do it. Moreover, in this project we not have additional funding’s to increase diapason of proteins that could be affected by our treatment and should be published also.
4. In Figures 6C and 7, the Western blot should be quantitative.
See also Supplementary file
5. The HPLC profile of Pseudevernia furfuracea should be included in this article.
Thank you for this suggestion, the HPLC, NMR analyses were published in our previous paper, as is citated in Material section. I addition, it is not correct to use the same results in two different articles.
Petrova, K., Kello, M., Kuruc, T., Backorova, M., Petrovova, E., Vilkova, M., ... & Mojzis, J. (2021). Potential Effect of Pseudevernia furfuracea (L.) Zopf Extract and Metabolite Physodic Acid on Tumour Microenvironment Modulation in MCF-10A Cells. Biomolecules, 11(3), 420.
Round 2
Reviewer 4 Report
accepted